# UMCI: A Unified Counterfactual Framework for Robust Vision-Language Reasoning

## Abstract

Integrating Large Language Models into vision-language frameworks has led to the rise of powerful Large Vision-Language Models (LVLMs). However, this integration introduces two critical robustness challenges: language bias and language sensitivity. To address these issues, we propose a **U**nified **M**ulti-round **C**ounterfactual **I**nference (UMCI) framework, which generalizes and extends prior methods like Counterfactual VQA and Visual Contrastive Decoding. UMCI performs multiple rounds of counterfactual inference using both textual and visual perturbations to mitigate bias and enhance consistency. This process reveals a novel test-time scaling law: increasing the number of counterfactual rounds consistently improves robustness. We also notice that non-robust samples are not fixed across different LVLMs. To disentangle the effects of the proposed inference algorithm from the confounding effect introduced by the base models, we introduce the dynamic Bias and Sensitivity Benchmark (BS Benchmark) as an adaptive evaluation tool specifically designed to probe robustness issues tailored to each LVLM. Our experiments demonstrate that UMCI significantly improves robustness on BS Benchmark while enhancing or at least maintaining the performance on standard benchmarks such as MMBench-CN/EN, MME, MMStar, CCBench, and ViLP. Extensive experimental results indicate that UMCI is scalable, generalizable, and offers a promising path toward robust multimodal reasoning.

## 1 Introduction

The recent advance in Large Language Models (Brown et al., 2020; Achiam et al., 2023; Touvron et al., 2023; Bai et al., 2023a; Liu et al., 2024a) (LLMs) has not only revolutionized the field of natural language processing but also catalyzed significant progress in multi-modal research, particularly in the vision-language domain (Yin et al., 2024; Zhang et al., 2024). To better utilize the knowledge of LLMs, the prevalent training framework for Large Vision-Language Model (LVLM) integrates a visual encoder with a pretrained LLM and jointly fine-tunes the combined architecture, resulting in powerful and versatility LVLMs such as InstructBLIP (Dai et al., 2023), LLaVA series (Liu et al., 2023; 2024b) and Qwen-VL series (Bai et al., 2023b; Wang et al., 2024).

However, these LVLMs continue to suffer from robustness issues in two key aspects. First, the above-mentioned LLM-based vision-language framework inevitably inherits certain drawbacks of LLMs, such as sensitivity to language prompts (Arora et al., 2023; Jiang et al., 2023; Wightman et al., 2023). Conventional VQA models lack the large-scale pretraining of LLMs and thus can only understand very limited textual information, failing to capture subtle prompt variations and thereby side-stepping this issue. As illustrated in Figure 1(a), simply requesting a LVLM to check image details without altering the question results in different outputs for the same input image. This language sensitivity undermines the consistency of LVLMs, reducing their reliability from the user's perspective. Second, vision-language models are also known to be susceptible to language bias. For example, conventional Visual Question Answering (VQA) models often rely heavily on language priors to answer questions, disregarding visual input (Niu et al., 2021; Wen et al., 2021). As shown in Figure 1(b), this problem also persists in LVLMs and can sometimes lead to generating non-existent contents, known as object hallucination (Li et al., 2024; Leng et al., 2024).

Recently, a growing number of research has focused on mitigating object hallucination in LVLMs (Zhou et al., 2024; Li et al., 2024). Among these efforts, Visual Contrastive Decoding

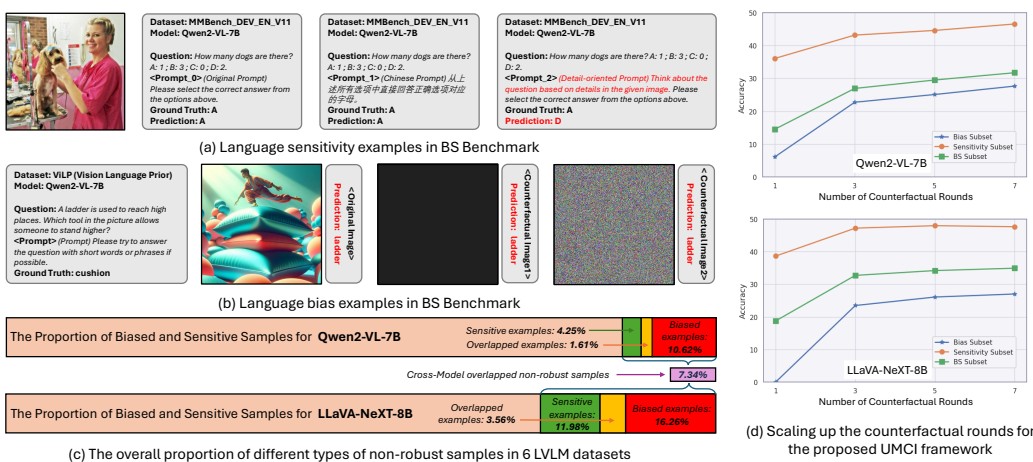

Figure 1: (a) and (b) are real BS Benchmark examples suffering from language sensitivity and bias issues; (c) shows the overall proportion of different types of non-robust samples across all 6 datasets under two commonly used LVLMs; (d) demonstrates a novel test-time scaling law of robustness regarding the increased counterfactual rounds in the proposed UMCI.

(VCD) (Leng et al., 2024) and its variants (Woo et al., 2024; Suo et al., 2025) have emerged as some of the most effective and widely adopted solutions. These methods typically perform a standard inference to obtain baseline logits and then estimate potential biases via a secondary inference with perturbed inputs. The final unbiased prediction is derived by weighted subtraction of the two logits. However, the object hallucination is merely a continuation of the language bias observed in conventional VLMs (Niu et al., 2021; Tang et al., 2020), and it ignores the issue of language sensitivity that is newly introduced by LVLMs.

In this work, we first analyze the underlying principles of VCD, particularly the role of the trade-off parameter $\alpha$, which is absent in the original Contrastive Decoding (CD) (Li et al., 2023). Through an in-depth mathematical analysis, we demonstrate that VCD is theoretically aligned with some debiasing algorithms used in previous vision-language tasks, such as TDE (Tang et al., 2020) and TIE (Niu et al., 2021). Specifically, VCD leverages TIE logits to reweight the original logits, where $1/\alpha$ acts as the temperature parameter for logit scaling. Building on this insight, we propose a more comprehensive inference framework, termed **U**nified **M**ulti-round **C**ounterfactual **I**nference **(UMCI)**, which unifies both Textual Counterfactual (TC) and Visual Counterfactual (VC) components. The final prediction is then derived from aggregating and comparing all multi-round counterfactual logits. This approach generalizes VCD and enables the simultaneous mitigation of both bias and sensitivity issues. We further examine three configurations: $UMCI_3$, $UMCI_5$, and $UMCI_7$, with different numbers of input variations to investigate the effect of increasing counterfactual inference rounds. We argue that our approach establishes a novel test-time scaling law, distinct from prior methods that increase intermediate token lengths within a single inference. Instead, robustness is enhanced by performing multiple rounds of counterfactual inference.

We also introduce a new evaluation benchmark, termed **B**ias and **S**ensitivity Benchmark **(BS Benchmark)**, to adaptively assess the robustness improvements of individual models. The key motivation behind BS Benchmark is that those non-robust data samples are not fixed. As shown in Figure 1(c), among all 24.68% hard samples for one LVLM(LLaVA-NeXT), there are only 7.34% shared with another LVLM(Qwen2-VL). This suggests that an LVLM may perform perfectly well on a fixed robustness dataset for previous models, yet still be vulnerable to other new samples. To enable a more precise analysis of algorithmic contributions, it is essential to disentangle the robustness gains from the confounding effect of base model performance. To this end, this benchmark is constructed by adaptively extracting non-robust subsets from existing LVLM datasets, based on the performance of a given LVLM. These model-specific subsets prevent newly introduced LVLMs from covering robustness issues by overfitting to existing datasets. Notably, the BS Benchmark is easily scalable and can be seamlessly applied to widely used real datasets such as MMBench, MME, *etc.*, introducing more diverse and nature question types than previous datasets (Li et al., 2024). Furthermore, as

illustrated in Figure 1(c), the additional statistical information itself from BS Benchmark facilitates a more comprehensive diagnosis of the inherent vulnerabilities of each LVLM.

The main contributions of this paper are threefold: 1) We propose UMCI, a unified counterfactual inference framework that simultaneously mitigates language bias and enforces language consistency. 2) We introduce the BS Benchmark, a model-specific and dynamic benchmark designed to better assess the robustness of LVLMs under samples from real downstream tasks. 3) We demonstrate that UMCI consistently improves performance on both the BS Benchmark and standard datasets, exhibiting strong generalizability. Furthermore, we uncover a novel test-time scaling law that links the number of counterfactual inference rounds to robustness gains.

## 2 RELATED WORK

**Large vision-language models.** LVLMs integrate two of the most significant breakthroughs in recent years: the versatile image encoder CLIP (Radford et al., 2021) and LLMs for general-purpose question answering (Radford et al., 2019; Touvron et al., 2023). The typical inference pipeline of a LVLM proceeds as follows: the input image is first encoded by CLIP or its more advanced successors (Zhai et al., 2023) to extract patch-level visual features; an adapter then maps these features to the token embedding space of the LLM (Liu et al., 2023; Bai et al., 2023b); finally, the visual and textual token embeddings are jointly fed into the LLM to generate the response. LVLMs have shown broad applicability in vision-language tasks such as image captioning (Xu et al., 2015; Yang et al., 2019) and Visual Question Answering (VQA) (Antol et al., 2015).

**Language bias and sensitivity in vision-language models.** Language bias has been a longstanding challenge for visual-language models. Previously, it was widely studied as the language prior in tasks like VQANiu et al. (2021); Goyal et al. (2017). In today's LVLMs, it commonly manifests as object hallucination. Recent works have sought to mitigate it through targeted retraining and contrastive decoding strategiesGunjal et al. (2024); Leng et al. (2024); Jiang et al. (2025), which are parallel to earlier techniques such as rebalanced training and counterfactual inference (Chen et al., 2020; Niu et al., 2021). Meanwhile, sensitivity to language prompts has received considerably less attention in VL research. Early VQA systems side-stepping this issue by using a small language encoder. The emergence of LLMs has brought it to the forefront. Existing mitigation strategies can be broadly categorized into three groups: 1) prompt ensembling (Pitis et al., 2023); 2) RL-based prompt optimization (Kwon et al., 2024); 3) Chain-of-thought verification (Wang et al., 2022).

**Test-time scaling laws.** Scaling laws have always been central to understanding LLM behavior, particularly the positive correlation between the scale of model/dataset/compute and the performance (Kaplan et al., 2020; Hoffmann et al., 2022). Recently, the attention has shifted toward test-time scaling, where increasing inference-time compute is also critical (Snell et al., 2025), such as adding demonstrations or decoding steps. In this work, we extend the notion of test-time scaling to the robustness: rather than increasing intermediate token length in a single inference, our method improves LVLM robustness by aggregating logits across more counterfactual inference rounds.

## 3 METHODOLOGY

### 3.1 PRELIMINARIES

**Counterfactual VQA:** the use of counterfactual inference to mitigate language bias in vision-language tasks dates back to Unbiased SGG (Tang et al., 2020) and CF-VQA (Niu et al., 2021). These works were the first to introduce the concepts of Total Direct Effect (TDE) and Total Indirect Effect (TIE) from the field of causality to achieve unbiased estimations via logit subtraction.

Since an LVLM can be regarded as a general VQA model, we take CF-VQA as an example. The TIE-based counterfactual logits can be formulated as:

$$TIE = Z(q, v, k) - Z(q, v^*, k^*), \tag{1}$$

where $Z(\cdot)$ denotes the model producing answer logits, $q$ denotes the question feature, $v$ is the visual feature, $k$ is the multi-modal fusion feature, $v^*$ and $k^*$ are counterfactual dummy features agnostic to the inputs. In conventional VQA, which is formulated as a closed-set classification task, the unbiased answer is obtained by returning the candidate answer with the highest TIE logits.

**Visual Contrastive Decoding (VCD):** building upon the idea of Contrastive Decoding (CD) (Li et al., 2023), VCD extends CD to mitigate object hallucination during LVLM inference, which can be formulated as follows:

$$p(y|v, v^*, q) = softmax((1 + \alpha)\,\text{logit}(y|v, q) - \alpha\,\text{logit}(y|v^*, q)), \tag{2}$$

where $y$ denotes the generated discrete token, $\alpha$ is a trade-off hyperparameter, $q$ and $v$ represent the input textual and visual tokens, respectively, and $v^*$ corresponds to visual tokens obtained from a noisy image. The previously generated tokens are considered part of $q$ for simplicity. The final VCD answer is therefore iteratively sampled from $p(y|v, v^*, q)$.

## 3.2 Unified Multi-round Counterfactual Inference

In this paper, we observe that VCD essentially reweights the original logits using TIE logits from CF-VQA. Building on this insight, we propose a Unified Multi-round Counterfactual Inference (UMCI) framework, which enhances model robustness through systematic logit-level reasoning over textual and visual counterfactual samples. The proposed UMCI framework not only unifies the formulations of VCD and CF-VQA, but also provides a principled solution to both language bias and sensitivity.

We begin by revisiting VCD through the lens of CF-VQA. Specifically, we treat object hallucination in LVLMs as the consequence of iterative biased token generation and frame the decoding process as a sequence of biased classifications. This perspective highlights that LVLMs are fundamentally no different from conventional VQA models. At each generation step, the bias can be mitigated through reasoning over counterfactual logits. Based on this observation, we transform the probability expression in equation 2 into a logit-based formulation $Z_{vcd}(v, v^*, q)$ as follows:

$$Z_{vcd}(v, v^*, q) = (1 + \alpha)\,Z(v, q) - \alpha\,Z(v^*, q), \tag{3}$$

where $Z(\cdot)$ denotes the LVLM that takes both textual tokens $q$ and visual tokens (either $v$ from real images or $v^*$ from dummy ones) as input and output the logits for the next token. Since there are no explicit multi-modal fusion features in the LVLM inputs, we removes $k$ or $k^*$ in original TIE.

To better understand the relationship between VCD and TIE, we transform the above VCD logits into the $\exp(\cdot)$ domain. By explicitly expanding the $softmax$ function $exp(x_i)/(\sum_j exp(x_j))$ and omitting the normalization term, we approximate the probability using $p(y) \propto exp(\cdot)$. With this simplification, the VCD probability $p(y|v, v^*, q)$ in equation 2 can be rewritten as:

$$\begin{aligned}
p(y|v, v^*, q) &= softmax(Z_{vcd}(v, v^*, q)) \\
&\propto exp(Z(v, q) + \alpha\,(Z(v, q) - Z(v^*, q))) \\
&= exp(Z(v, q)) \cdot exp(\alpha\,(Z(v, q) - Z(v^*, q))) \\
&= exp(Z(v, q)) \cdot exp(TIE/\tau). \tag{4}
\end{aligned}$$

The above formulation bridges VCD and CF-VQA, showing that VCD essentially performs weighted token generation upon the original output token probability $p(y|v, q) \propto exp(Z(v, q))$, where TIE logits $exp(TIE/\tau)$ serves as a vocabulary-wise reweighting term, thus forcing the model to rely on visual difference. This formulation also clarifies the role of $\alpha$ in VCD. Neither vanilla CD nor TIE itself requires this additional parameter, because the logit difference itself captures the useful effect of real $v$ over dummy $v^*$. Yet, as a reweighting term, it requires a temperature scaling factor to adjust the trade-off strength, so we further denote $\tau = 1/\alpha$.

To establish a more general robust inference framework, it is also necessary to address the overlooked language sensitivity issue as well. Therefore, we propose UMCI framework to incorporates both a Visual Counterfactual (VC) component, which enhances visual cues similar to TIE, and a Textual Counterfactual (TC) component, which ensures prompt-consistent logits, as follows:

$$p_{\text{UMCI}}(y|\boldsymbol{v}, \boldsymbol{q}) \propto exp(\text{TC}/\tau_1) \cdot exp(\text{VC}/\tau_2), \tag{5}$$

$$\text{TC}_k = max_i(Z_k(v^0, q^i)),\ i = 0, 1, 2, ..., N \tag{6}$$

$$\text{VC} = Z(v^0, q^0) - \mathbb{E}[Z(v^j, q^0)],\ j = 1, 2, ..., M \tag{7}$$

where $\boldsymbol{v} = \{v_j\}_{j=0}^M$ and $\boldsymbol{q} = \{q_i\}_{i=0}^N$ denote overall inputs; $M$ and $N$ are the number of visual and textual counterfactual variations, respectively; $v^0$ and $q^0$ stand for original visual and textual

tokens; $\{v^j, j \neq 0\}$ and $\{q^i, i \neq 0\}$ represent counterfactual visual tokens generated from content-removed images and counterfactual textual tokens from semantically equivalent but lexically different prompts, respectively. The detailed implementation of these counterfactual samples will be explained in Experiments and Appendix D. The operator $max_i(Z_k(\cdot))$ computes the element-wise maximum over $N + 1$ samples on the $k-$th dimension of the logits for better consistency. VC enhances the original TIE by incorporating multiple counterfactual visual inputs to obtain a more stable estimation. $\tau_1$ and $\tau_2$ are temperature scaling factor for TC and VC logits, respectively. Following VCD, we also adopt Adaptive Plausibility Constraints as a post process before sampling from $p_{\text{UMCI}}(y)$, details will be given in Appendix C.

The overall UMCI framework provides a generalized solution for robust LVLM inference. In this unified framework, prior works such as VCD and CF-VQA can be viewed as special cases. For VCD, there are no counterfactual prompt variations ($N = 0$) and only one counterfactual image ($M = 1$). For CF-VQA, the entire TC component is set to a constant and $M = 1$. As demonstrated in our experiments, increasing the number of counterfactual inference rounds, *i.e.* using larger $M$ and $N$, leads to more robust final outputs, revealing an emergent test-time scaling law for robustness in LVLMs. We also believe that there remains a large opportunity to improve the effectiveness by developing more advanced TC and VC algorithms in future work.

### 3.3 BIAS AND SENSITIVITY BENCHMARK

Collecting and constructing datasets tailored to specific robust issues is often cumbersome and costly. What's worse, once such datasets are publicly released, they may be inadvertently integrated into the web-crawled training corpus of subsequent LVLMs. To better evaluate language bias and sensitivity in real downstream tasks, we introduce the Bias and Sensitivity Benchmark (BS Benchmark), guided by two main motivations: 1) the evaluation benchmark should be model-specific and dynamic. Since different LVLMs may exhibit varying levels of robustness and their vulnerable samples are not the same, it is important to disentangle the confounding effect of the base model performance

| Subset Size | B Subset | S Subset | BS Subset | Overlap |
|---|---|---|---|---|
| LLaVA-NeXT (MCQ) | 1810 | 1005 | 2476 | 339 |
| LLaVA-NeXT (Others) | 345 | 582 | 794 | 133 |
| LLaVA-NeXT (Overall) | 2155 | 1587 | 3270 | 472 |
| Qwen2-VL (MCQ) | 1080 | 252 | 1243 | 89 |
| Qwen2-VL (Others) | 327 | 311 | 513 | 125 |
| Qwen2-VL (Overall) | 1407 | 563 | 1756 | 214 |

Table 1: The size of each subset in constructed BS Benchmark. The overall number of samples across all 6 datasets is 13251, with MCQ and Others categories being 10632 and 2619, respectively.

from the improvements brought by different inference strategies, so we can better understand the contribution of the inference algorithm itself; 2) existing LVLM bias evaluation datasets typically focus on a single question type and adopt formats that differ significantly from real-world LVLM tasks, *e.g.* exist-or-not questions (Yifan et al., 2023). Therefore, it is necessary to develop methods that can automatically adapt to diverse question types and task formats.

Following the above two guiding principles, the proposed benchmark enables the transformation of any popular or newly released LVLM dataset regardless of its question formats into a robustness evaluation benchmark. Specifically, it will adaptively generate model-specific bias subset, sensitivity subset and their union BS Subset for any given LVLM dataset through a two-step process. First, we will evaluate the dataset using the given model. Then, we will adopt the following criteria for filtering the Bias Subset (BS) and the Sensitivity Subset (SS):

$$\text{BS} = \{(a_{gt}, v^0, q^0) \mid \forall j \neq 0, \arg\max_a p(a|v^0, q^0) = \arg\max_a p(a|v^j, q^0) \neq a_{gt}\}, \quad (8)$$

$$\text{SS} = \{(a_{gt}, v^0, q^0) \mid \forall i \neq 0, \arg\max_a p(a|v^0, q^0) \neq \arg\max_a p(a|v^0, q^i)\}, \quad (9)$$

where $a$ and $a_{gt}$ denote the predicted answer and the ground-truth answer, respectively, and $\arg\max_a p(a|\cdot)$ means the predicted answer is obtained via greedy decoding. The generation of counterfactual inputs $v^j$ and $q^i$ follows the same procedure as in UMCI. In this paper, we fix $M = N = 2$ for all our subsets construction. In essence, for BS (equation 8), we select samples that yield the same incorrect predictions under both the original and dummy visual inputs, indicating a reliance on spurious language priors; for SS (equation 9), we identify samples whose predictions change in response to subtle, non-causal prompt variations. The final BS Subset is defined as the union of the above two subsets, enabling the investigation of both bias and sensitivity

issues. We further split all samples into two groups based on their question types: MCQ for the dominant Multiple-Choice Question type and Others for Yes/No or general open-ended QA types.

In summary, the proposed BS Benchmark offers three key advantages. First, robustness is a model-specific problem, samples that are biased or sensitive for one model may not be vulnerable for another, more evidence will be provided in Table 3. An adaptive and model-specific robustness benchmark can thus prevent newly developed LVLMs from being exposed to publicly released fixed datasets and misleading the evaluation of their real underline robustness. Second, as shown in Figure 1(c), different models exhibit varying levels of robustness, the size of each subset provides valuable insight into different models. For example, Table 1 indicates that: 1) Qwen2-VL is generally more robust than LLaVA-Next; 2) Qwen2-VL is more vulnerable to bias than to sensitivity; and 3) LLaVA-NeXT exhibits more sensitivity issues compared to Qwen2-VL. Third, BS Benchmark enables the evaluation of robustness in various real-world tasks, rather than predefined questions such as a simple exist-or-not (Yes/No) assessment commonly used in previous work (Yifan et al., 2023). It also allows for the effortless conversion of any real-world LVLM dataset into the BS Benchmark format, eliminating the need for labor-intensive sample collection and manual annotation.

## 4 EXPERIMENTS

### 4.1 BENCHMARK SETTINGS

In our experiments, we construct BS Benchmark using 6 widely adopted LVLM benchmarks: MME (Fu et al., 2023), MMStar (Chen et al., 2024), CCBench (Liu et al., 2024c), ViLP (Luo et al., 2024), MMBench-DEV-EN-V11 and MMBench-DEV-CN-V11 (Liu et al., 2024c). We begin by randomly splitting the datasets into 20% validation and 80% test sets, resulting in 3315 and 13251 samples, respectively. Detailed subset statistics are provided in Table 1. Note that the size of BS Benchmark increases with larger number of $M$ and $N$. For consistency and convenience, we fix $M = N = 2$ for all subsets constructions throughout our experiments. As we mentioned, to enable a more fine-grained analysis, we separately report performance for Multiple-Choice Question (MCQ) and Others (Open-ended QA for ViLP or Yes/No for MME) categories, in addition to the overall results. We use top-1 accuracy as the evaluation metric for all experiments. For the MME dataset, which adopts a different scoring metric, we convert its results to accuracy, so they can be integrated with samples from other datasets to get the final results.

### 4.2 IMPLEMENTATION DETAILS

**Environments and Model Zoo.** All experiments were conducted using VLMEvalKit (Duan et al., 2024) on a single NVIDIA A800 GPU (80GB) with environment: Pytorch=2.6, Transformers=4.49, and Flash Attention=2.7 (Dao, 2023). We used Hugging Face versions of Qwen2-VL-7B-Instruct (Wang et al., 2024) and Llama3-LLaVa-NeXT-8B-hf (Liu et al., 2024b) as our base models. Following their default configurations, the experiments were conducted using bfloat16 precision and top-k sampling decoding for Qwen2-VL, while LLaVa-NeXT used float16 and greedy decoding.

**Algorithm details.** We evaluated 4 inference strategies: TIE, VCD, M3ID, and the proposed UMCI. We adapted Total Indirect Effect (TIE) from CF-VQA (Niu et al., 2021) to LVLMs by removing the multi-modal features $k$ and $k^*$ in Eq. equation 1. For fair comparison, we also incorporated the Adaptive Plausibility Constraints used in VCD and M3ID into TIE. VCD (Leng et al., 2024) and M3ID (Favero et al., 2024) share the same mathematical formulation as Eq. equation 4, except that the hyperparameter $\tau$ in M3ID varies depending on the position of the predicted token. For the proposed UMCI, we added subscripts such as $UMCI_3$, $UMCI_5$, and $UMCI_7$ to indicate the number of inference rounds. For example, $UMCI_5$ means that the total number of counterfactual visual and textual variations, together with the original inputs is 5, *i.e.*, $M + N + 1 = 5$ In our experiments, we set $M = N = 1$, $M = N = 2$, and $M = N = 3$ for $UMCI_3$, $UMCI_5$, and $UMCI_7$, respectively.

**Counterfactual sample construction.** We constructed up to 3 visual counterfactual variations and 3 prompt variations: 1) VC-Color0(C0) renders the input image into black; 2) VC-Noise500(N500) and 3) VC-Noise400 apply the diffusion noise function from VCD, using noise steps of 500 and 400, respectively; 3) TC-V1 adds an additional system prompt instructing the model to focus on image details; 4) TC-V2 further modifies the system prompt's language from English to Chinese or

| Method | B Subset | | | S Subset | | | BS Subset | | |
|---|---|---|---|---|---|---|---|---|---|
| | MCQ | Others | Overall | MCQ | Others | Overall | MCQ | Others | Overall |
| LLaVA-NeXT | 0.0 | 0.0 | 0.0 | 39.2 | 37.63 | 38.63 | 15.91 | 27.58 | 18.75 |
| LLaVA-NeXT-TIE | 12.98 | 23.48 | 14.66 | 39.00 | 57.56 | 45.81 | 21.89 | 44.21 | 27.31 |
| LLaVA-NeXT-VCD | 12.65 | 25.51 | 14.71 | 40.50 | 56.53 | 46.38 | 22.54 | 44.58 | 27.89 |
| LLaVA-NeXT-M3ID | 16.91 | 25.22 | 18.24 | 39.90 | 56.36 | 45.94 | 24.15 | 44.33 | 29.05 |
| LLaVA-NeXT-UMCI$_3$ (ours) | 21.22 | 35.36 | 23.48 | 39.60 | 60.31 | 47.20 | 27.14 | 50.13 | 32.72 |
| LLaVA-NeXT-UMCI$_5$ (ours) | 23.81 | 37.97 | 26.08 | 40.60 | 60.65 | 47.95 | 28.80 | 51.01 | 34.19 |
| LLaVA-NeXT-UMCI$_7$ (ours) | 24.86 | 38.26 | 27.01 | 40.10 | 60.65 | 47.64 | 29.68 | 51.26 | 34.92 |
| Qwen2-VL | 5.37 | 8.56 | 6.11 | 38.10 | 34.41 | 36.06 | 10.78 | 23.59 | 14.52 |
| Qwen2-VL-TIE | 16.20 | 16.82 | 16.35 | 45.63 | 36.66 | 40.67 | 20.27 | 27.29 | 22.32 |
| Qwen2-VL-VCD | 15.74 | 21.71 | 17.13 | 46.83 | 40.84 | 43.52 | 20.11 | 30.41 | 23.12 |
| Qwen2-VL-M3ID | 19.81 | 21.71 | 20.26 | 47.22 | 41.16 | 43.87 | 23.65 | 30.6 | 25.68 |
| Qwen2-VL-UMCI$_3$ (ours) | 21.67 | 26.30 | 22.74 | 44.05 | 42.44 | 43.16 | 24.54 | 32.75 | 26.94 |
| Qwen2-VL-UMCI$_5$ (ours) | 24.91 | 25.69 | 25.09 | 47.22 | 42.44 | 44.58 | 28.00 | 33.14 | 29.50 |
| Qwen2-VL-UMCI$_7$ (ours) | 27.04 | 29.66 | 27.65 | 47.22 | 45.98 | 46.54 | 29.61 | 36.84 | 31.72 |

Table 2: Experiments on B(ias) Subset, S(ensitivity) Subset, and BS Subset. **Bold texts** indicate the best result of each column and underline texts indicate the second best result.

vice versa; 5) TC-V3 that injects identity information by prompting the model to respond as a clever student. More detailed prompts will be given in the Appendix D.

**Hyperparameter settings.** Based on the validation results, we set $\tau_1$ to 1.5, 2, and 2.5 for UMCI$_3$, UMCI$_5$, and UMCI$_7$, respectively. Since the TC component involves element-wise maximum over logits, its magnitude increases with the number of variations $N$. Therefore, the temperature scaling factor $\tau_1$ should be increased accordingly to maintain a similar distribution of TC logits. The $\tau_2$ is fixed at 0.2, because the averaging operation in the VC logits stabilizes the distribution and mitigates

| Construction Model | Methods | MCQ | Others | Overall |
|---|---|---|---|---|
| LLaVA-NeXT | LLaVA-NeXT-Original | 15.91 | 27.58 | 18.75 |
| | LLaVA-NeXT-UMCI$_5$ | 28.80 | 51.01 | 34.19 |
| | Qwen2-VL-Original | 59.29 | 63.48 | 60.31 |
| | Qwen2-VL-UMCI$_5$ | 61.15 | 67.88 | 62.78 |
| Qwen2-VL | Qwen2-VL-Original | 10.78 | 23.59 | 14.52 |
| | Qwen2-VL-UMCI$_5$ | 28.00 | 33.14 | 29.50 |
| | LLaVA-NeXT-Original | 30.25 | 39.18 | 32.86 |
| | LLaVA-NeXT-UMCI$_5$ | 34.59 | 41.33 | 36.56 |

Table 3: Ablation on cross-model BS Subset evaluation.

the need for the scaling change. For the Adaptive Plausibility Constraint (Leng et al., 2024) used in our experiments, the threshold parameter is set to 0.3 unless otherwise specified. More details about the constraint and hyperparameter ablation will be introduced in the Appendix E.

## 4.3 EXPERIMENTAL RESULTS

**Experiments on the proposed BS Benchmark.** As shown in Table 2, we adopted two state-of-the-art LVLMs for our experiments: LLaVA-NeXT-8B and Qwen2-VL-7B. We compared the base model performances and three other algorithms: TIE, VCD, and M3ID that utilized counterfactual inference. The proposed methods, UMCI$_3$, UMCI$_5$, and UMCI$_7$, consistently demonstrated superior performance across the B(ias), S(ensitivity), and combined BS Subsets. We further reported MCQ and Others results based on question types and saw that the improvements brought by UMCI were consistent across both categories. Table 2 also reveals that the proposed BS Benchmark can successfully disentangle the base model performance and focus on investigating the effectiveness of inference algorithms, as Qwen2-VL outperforms LLaVA-NeXT by 10.19% on the original datasets in Table 4, while their base and final overall performances on the BS Benchmark are very close.

**Experiments on real-world LVLM datasets.** We further evaluated the proposed UMCI on 6 popular LVLM datasets to verify its performance under real-world data distributions, in addition to the proposed subsets alone. Taking UMCI$_5$ as an example in Table 4, it consistently outperformed the baseline models in all question types and almost all datasets. Meanwhile, TIE, VCD and M3ID decrease the performance on Others question type. Note that although the improvements appear

| Method | Single Dataset | | | | | | Gathered by Question Type | | |
|---|---|---|---|---|---|---|---|---|---|
| | MMB-C | MMB-E | MME | CCB | MMS | ViLP | MCQ | Others | Overall |
| LLaVA-NeXT | 78.0 | 79.72 | 79.57 | 47.0 | 44.75 | 51.53 | 70.12 | 71.86 | 70.46 |
| LLaVA-NeXT-TIE | 78.28 | 80.28 | 77.30 | 45.65 | 46.00 | 53.19 | 70.36 | 70.68 | 70.42 |
| LLaVA-NeXT-VCD | 78.38 | 80.28 | 78.09 | 46.63 | 45.00 | 54.31 | 70.44 | 71.55 | 70.66 |
| LLaVA-NeXT-M3ID | 78.31 | 80.18 | 78.62 | 45.89 | 45.92 | 54.03 | 70.36 | 71.86 | 70.66 |
| LLaVA-NeXT-UMCI$_5$ (ours) | 78.21 | 80.08 | 80.15 | 46.20 | 45.75 | 53.06 | 70.32 | 72.70 | 70.79 |
| Qwen2-VL | 85.26 | 86.36 | 87.89 | 73.22 | 59.50 | 56.53 | 80.91 | 79.27 | 80.58 |
| Qwen2-VL-TIE | 86.00 | 86.59 | 86.52 | 73.84 | 59.00 | 57.19 | 81.30 | 78.43 | 80.73 |
| Qwen2-VL-VCD | 86.05 | 86.56 | 86.41 | 73.77 | 60.08 | 57.92 | 81.42 | 78.58 | 80.86 |
| Qwen2-VL-M3ID | 85.69 | 86.46 | 86.10 | 73.96 | 59.75 | 57.78 | 81.25 | 78.31 | 80.67 |
| Qwen2-VL-UMCI$_5$ (ours) | 85.97 | 86.67 | 87.36 | 73.59 | 59.92 | 58.06 | 81.39 | 79.31 | 80.98 |

Table 4: Experiments on MMB(ench-Dev)-C/E(N-V11), MME, CCB(ench), MMS(tar), and ViLP indicate that **UMCI has more consistent improvement** than TDE/VCD/M3ID on those real-world LVLM benchmarks (using 80% test splits). Blue texts indicate an improvement over the baseline.

relatively marginal, since vulnerable samples comprise only a portion of the datasets. These results confirm that the gains observed on BS Benchmark are not due to overfitting to specific data distributions, but rather reflect a general improvement in robustness.

**Ablation study on test-time scaling with increasing inference rounds.** To better understand the effect of each component in UMCI framework, we conducted an ablation study on UMCI$_5$. As shown in Table 5, we first evaluated the performance of the base inputs and four individual counterfactual inputs on the BS Subset. We then incrementally increased the number of counterfactual rounds to form progressively more complete versions of UMCI to reach UMCI$_5$. Note that experiments on VC component and TC component alone are also included. Together with the comprehensive results of UMCI$_3$, UMCI$_5$, and UMCI$_7$ in Figure 2, the overall findings highlight the significance of test-time scaling: robustness of models can be improved with more incorporated counterfactual rounds.

| Base | VC-C | VC-N | TC-V1 | TC-V2 | MCQ | Others | Overall |
|---|---|---|---|---|---|---|---|
| ✓ | | | | | 10.78 | 23.59 | 14.52 |
| | ✓ | | | | 8.77 | 18.52 | 11.62 |
| | | ✓ | | | 10.62 | 25.15 | 14.86 |
| | | | ✓ | | 10.38 | 24.37 | 14.46 |
| | | | | ✓ | 12.07 | 23.00 | 15.26 |
| ✓ | ✓ | | | | 21.72 | 29.43 | 23.97 |
| ✓ | | | ✓ | | 10.54 | 23.39 | 14.29 |
| ✓ | ✓ | | ✓ | | 24.54 | 32.75 | 26.94 |
| ✓ | | ✓ | | ✓ | 26.67 | 30.97 | 27.93 |
| ✓ | ✓ | ✓ | | | 27.37 | 31.13 | 28.46 |
| ✓ | | | ✓ | ✓ | 11.58 | 23.21 | 14.98 |
| ✓ | ✓ | | ✓ | ✓ | 26.07 | 32.55 | 27.96 |
| ✓ | ✓ | ✓ | ✓ | | 26.71 | 33.33 | 28.64 |
| ✓ | ✓ | ✓ | ✓ | ✓ | 28.00 | 33.14 | 29.50 |

Table 5: Ablation experiments for different counterfactual logits combinations using Qwen2-VL on BS Subset.

**Ablation study on cross-model BS Benchmark evaluation.** The ablation study in Table 3 provides additional insights: 1) non-robust samples vary significantly across different LVLMs. For instance, the BS Subset constructed by LLaVA-NeXT yields only 18.75% accuracy on its own model, while Qwen2-VL achieves 60.31% accuracy on the same subset, and vice versa. This demonstrates that even if an LVLM performs perfectly well on a fixed robustness benchmark, it may still fail on new vulnerable samples. These findings highlight the necessity of adopting a model-specific BS Benchmark; 2) The performance gains achieved through UMCI in one model are transferable to BS benchmarks constructed by other models, thereby validating the generalization ability of UMCI.

## 4.4 DISCUSSIONS

We also provide some interesting discussions to shed lights on the proposed UMCI framework and BS Benchmark.

**Q1: Why did the base models perform so poorly (*e.g.*, LLaVA-NeXT even got 0.0 on the Bias Subset) on the Bias, Sensitivity, and BS Subsets? A1:** The proposed BS Benchmark are intentionally designed to probe samples particularly vulnerable to robust issues, *i.e.* they are hard examples

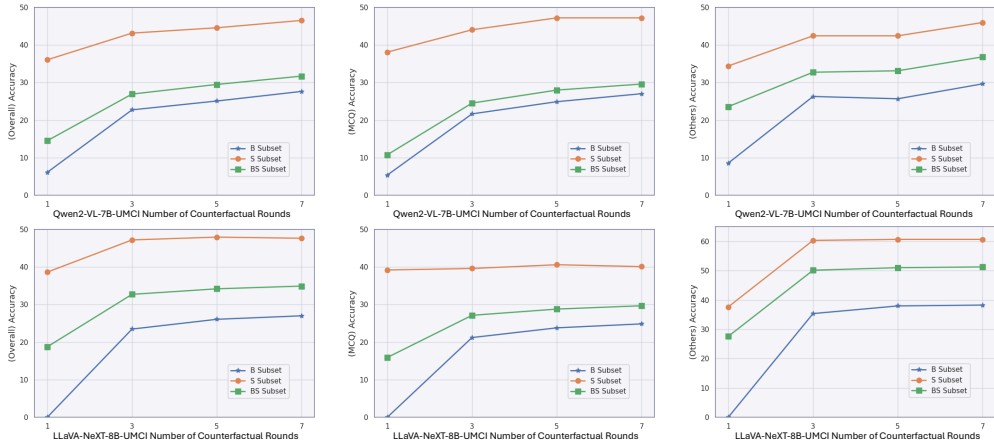

Figure 2: Investigating the test-time scaling law of robustness with respect to the number of inference rounds on B/S/BS subsets across different question types and LVLMs.

for LVLMs. That's why the model performances on these subsets are sometimes even lower than random guessing, *e.g.*, MCQs have a 25% chance accuracy for random guess. In fact, as defined in equation 8, the Bias Subset specifically collects samples for which the base model consistently produces incorrect predictions, so its accuracy is theoretically expected to be 0.0. The reason why Qwen2-VL does not yield exactly 0.0 is due to its use of top-k sampling for decoding by default, which introduces randomness into its outputs. In contrast, LLaVA-NeXT uses greedy decoding, producing deterministic predictions, which explains its consistent 0.0 accuracy on the Bias Subset.

**Q2: What's the computational overhead of UMCI and are there potential solutions for acceleration? A2:** The test-time scaling law entails a trade-off between inference time and performance, which means that the proposed UMCI will inevitably take more time. The most intuitive acceleration method for UMCI is batch inference. Based on our experiments, the computational overhead of $UMCI_3$, $UMCI_5$, and $UMCI_7$ using batch inference is approximately $1.29\times$, $1.81\times$, and $2.48\times$ that of the base model, respectively, which is much faster than the vanilla version, which costs $2.96\times$, $5.01\times$, and $6.68\times$, respectively. We also believe that KV Cache sharing for the visual and textual tokens that remain unchanged is a potential acceleration technique for UMCI.

**Q3: Why is UMCI different from previous test-time scaling laws, and could it open up a new paradigm? A3:** Most of the existing test-time scaling studies (Snell et al., 2025) focus on increasing the length of intermediate thinking tokens, which remains limited to the prompt or language level. However, the prompt-level improvement only reveals whether the answer is correct or wrong. It provides no insight into whether, for instance, the input image increases the logit magnitude for a specific token such as "cushion". By introducing UMCI, we go beyond discrete token outputs and analyze the underlying continuous logit distributions through comparison and aggregation of counterfactual logits. This approach provides significantly richer information than simply using final predicted tokens. Therefore, we believe that UMCI opens up a promising new direction for test-time scaling laws, which utilizes richer output information.

## 5 CONCLUSION

This paper introduces the Multi-round Counterfactual Inference (UMCI), a generalized framework for robust inference in LVLMs that simultaneously addresses language bias and sensitivity issues through comprehensive logit-level counterfactual reasoning. Together with the proposed Bias and Sensitivity Benchmark, we provide both a methodology advancement and an adaptive evaluation for enhancing LVLM robustness on real-world datasets. Extensive experiments further demonstrate a scalable path towards better test-time robustness: by simply increasing the number of counterfactual rounds during inference and integrating more advanced logit-level reasoning algorithms. We hope that UMCI and BS Benchmark will serve as foundational paradigms and diagnostic standards to guide future research toward reliable LVLMs.

## REPRODUCIBILITY STATEMENT

To promote reproducibility, all code, data, and evaluation scripts of this paper will be publicly released upon publication. The original 6 datasets, the VLMEvalKit codebase, and the LVLM checkpoints used in this paper are sourced from publicly available repositories on GitHub or Hugging Face. We also provide detailed information on the hardware, software, and their respective versions in the Implementation Details section of the main paper. All hyperparameters are included in the main paper or appendix. We believe that the open-source spirit is a driving force for progress in the AI field, and, as such, we are committed to ensuring that our work can be easily reproduced by future researchers.

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

# A APPENDIX

The following appendix contains supplementary details and experimental results excluded from the main paper due to space constraints. The overall appendix includes: B) The Use of Large Language Models (as requested by the ICLR 2026 conference author guide); C) adaptive plausibility constraint; D) generation of counterfactual inputs; E) additional experimental results and analyses.

# B THE USE OF LARGE LANGUAGE MODELS (LLMS)

In this research, we have utilized Large Language Models (LLMs) solely for the purpose of text refinement and editing. The LLM was only used to improve overall readability. It **did NOT** contribute to the idea creation, code writing, generation of initial drafts, or figure creation.

We affirm that all research ideas, experimental design, and data analysis were developed by the human authors. We take full responsibility for all aspects of the manuscript and ensure that **NO** content generated by the LLM constitutes plagiarism or scientific misconduct.

# C ADAPTIVE PLAUSIBILITY CONSTRAINT

As mentioned in the main paper, we adopt adaptive plausibility constraint from VCD (Leng et al., 2024) and M3ID (Favero et al., 2024) as a post-processing step before sampling output tokens. This constraint masks tokens with low logit values under the original input, ensuring that low-confidence tokens are not sampled as final outputs. Specifically, the constraint can be formulated as:

$$Z_{vcd}(v, v^*, q)_k = -\infty, \tag{10}$$
$$\text{s.t. } Z(v, q)_k < \max_k(Z(v, q)) + \log(\beta), \tag{11}$$

where $k$ is the token index for logits; the logit with value $-\infty$ ensures that $p_{vcd}(y|v, v^*, q)_k = 0$ for the masked tokens; $\beta$ is the threshold; $\max_k(Z(v, q))$ is the largest logit value for original inputs.

The rationale behind the Adaptive Plausibility Constraint is that, although the output distribution under the original input may be biased, it can still serve as a valid filter to identify plausible candidate tokens. Only tokens with logits greater than $\max_k(Z(v, q)) + \log(\beta)$ are allowed to receive VCD logits and participate in final sampling. In contrast, low-confidence candidates with insufficient logits are directly masked out. As shown in Table 6, removing the adaptive plausibility constraint leads to a performance drop for $UMCI_5$ on the B/S/BS subsets, and results in an even greater performance degradation on the original datasets as we expected.

| Methods | Constraint | Original | B Subset | S Subset | BS Subset |
|---|---|---|---|---|---|
| Qwen2-VL | NA | 81.12 | 6.10 | 37.59 | 15.46 |
| Qwen2-VL-UMCI$_5$ | ✗ | 68.93 | 26.16 | 34.04 | 27.63 |
| Qwen2-VL-UMCI$_5$ | ✓ | 81.03 | 29.65 | 40.43 | 32.55 |

Table 6: Ablation study for the adaptive plausibility constraint. To evaluate the effect of adaptive plausibility constraint, we conducted experiments on validation sets of original 6 datasets together with B(ias)/S(ensitive)/BS Subsets.

| Method | Qwen2-VL | Qwen2-VL-UMCI$_3$ | Qwen2-VL-UMCI$_5$ | Qwen2-VL-UMCI$_7$ |
|---|---|---|---|---|
| Inference Time (w/o batch inference) | 540.47ms | 1599.65ms | 2707.16ms | 3611.18ms |
| Inference Time (w/ batch inference) | 540.47ms | 697.24ms | 978.14ms | 1342.86ms |

Table 7: We report the average inference time per sample on the MMStar dataset using one A800 GPU to illustrate the computational overhead introduced by UMCI. Note that the baseline speed w/o batch inference sequentially conduct each counterfactual inference round, while w/ batch inference, all counterfactual inference rounds are conducted in one batch. Therefore, the later is significantly faster than the baseline speed.

For the proposed Unified Multi-round Counterfactual Inference (UMCI) framework, we slightly change the constraint as follows:

$$p_{\text{UMCI}}(y|\boldsymbol{v}, \boldsymbol{q}) = 0, \tag{12}$$

$$\text{s.t. } TC_k/\tau_1 < \max_k(TC/\tau_1) + \log(\beta), \tag{13}$$

where the key difference is that we use Textual Counterfactual (TC) logits, scaled by a temperature factor, to replace the original logits as the masking criterion, as we believe TC provides more consistent predictions. The final output tokens are then sampled from the unmasked candidates with non-zero probabilities.

In our experiments, the default threshold $\beta$ is set to 0.3 following the previous paper (Favero et al., 2024) for all BS Benchmark experiments. We consider $\beta$ as a trade-off parameter between relying on de-biased logits and original logits. When $\beta$ approaches 1.0, the final output token closely resembles that produced by the original inputs. In contrast, when $\beta$ approaches 0.0, the constraint becomes negligible, and the output behaves as if no filtering is applied. For experiments on original LVLM datasets, we increase $\beta$ by 0.5 to 0.8, as these datasets exhibit less bias and the outputs are generally closer to those produced by the original inputs.

## D    GENERATION OF COUNTERFACTUAL INPUTS

In this section, we provide further details on the generation of counterfactual inputs. For the Visual Counterfactual input VC-Color0, we directly set the RGB values of all pixels in the input image to (0, 0, 0), resulting in a completely black image. For VC-Noise400 and VC-Noise500, we follow the method used in VCD (Leng et al., 2024), where Gaussian noise is added to simulate the forward diffusion process (Ho et al., 2020) at 400 and 500 time steps, respectively. The mathematical formulation of this forward process is as follows:

$$v_t = \sqrt{\bar{\alpha}_t} \cdot v_0 + \sqrt{1 - \bar{\alpha}_t} \cdot \epsilon, \tag{14}$$

where $v_t$ is the final noise image at at step $t$; $v_0$ is original image; $\epsilon \sim \mathcal{N}(0, 1)$ is random Gaussian noise; $\bar{\alpha}_t$ is cumulative product. The detailed implementation is available in the official GitHub repository of VCD.

For Textual Counterfactual input TC-V1, TC-V2, and TC-V3, as we can see from Figure 3, Figure 4, and Figure 5, each variations provide a semantically equivalent but lexically different prompts. Without change the meaning of instruction, TC-V1 adds an additional system prompt instructing the model to focus on image details, TC-V2 further modifies the system prompt's language from English to Chinese or vice versa, TC-V3 injects identity information by prompting the model to respond as a clever student.

| Methods | Hyperparameters | MCQ | Others | Overall |
|---|---|---|---|---|
| Qwen2-VL | - | 11.97 | 22.38 | 15.46 |
| Qwen2-VL-UMCI$_5$ | $\tau_1 = 2.0 \; \tau_2 = 0.2$ | 33.45 | 30.77 | 32.55 |
| Qwen2-VL-UMCI$_5$ | $\tau_1 = 2.0 \; \tau_2 = 2.0$ | 22.89 | 27.97 | 24.59 |
| Qwen2-VL-UMCI$_5$ | $\tau_1 = 2.0 \; \tau_2 = 1.0$ | 26.06 | 29.37 | 27.17 |
| Qwen2-VL-UMCI$_5$ | $\tau_1 = 2.0 \; \tau_2 = 0.5$ | 32.39 | 30.77 | 31.85 |
| Qwen2-VL-UMCI$_5$ | $\tau_1 = 20 \; \tau_2 = 0.2$ | 3.87 | 18.88 | 8.89 |
| Qwen2-VL-UMCI$_5$ | $\tau_1 = 10 \; \tau_2 = 0.2$ | 23.59 | 23.77 | 23.65 |
| Qwen2-VL-UMCI$_5$ | $\tau_1 = 1.0 \; \tau_2 = 0.2$ | 28.17 | 26.57 | 27.63 |
| Qwen2-VL-UMCI$_5$ | $\tau_1 = 0.2 \; \tau_2 = 0.2$ | 11.97 | 20.97 | 14.99 |

Table 8: Ablation study for temperature scaling hyperparameters $\tau_1$ and $\tau_2$ of UMCI. Experiments are conducted under validation set of BS Subset.

## E  ADDITIONAL EXPERIMENTS

This section will discuss some additional experiments, including ablation studies on hyperparameters, analysis of inference time for UMCI, and other supplementary results.

**Ablation study for hyperparameters.** As shown in Table 8, we select the temperature scaling hyperparameters for the TC and VC logits based on validation performance on the BS Subset. For fair comparison, the hyperparameters were select on UMCI$_5$ under base model Qwen2-VL and directly apply to LLaVA-NeXT. The temperature scaling $\tau_2$ for VC is fixed as 0.2 across UMCI$_3$, UMCI$_5$, and UMCI$_7$, because the logits distribution of VC would not change with the number of visual counterfactual inputs. As to the temperature scaling $\tau_1$ for TC, since the calculation of TC involves maximum cross all outputs using different textual counterfactual inputs, the logits distribution of TC would change with number of textual counterfactual variations. Therefore, we decide to intuitively add 0.5 to $\tau_1$ to prevent the distribution change when there is one more textual variation added to UMCI.

**Inference time and discussion about acceleration techniques.** As shown in Table 7, we first evaluate the computational overhead of the vanilla implementation (sequential counterfactual inference) of UMCI by measuring the average inference time per sample on the validation set of MMStar (Qwen2-VL BS Subset) using a single A800 GPU with Flash Attention 2.7. Specifically, we compare the original inference with UMCI$_3$, UMCI$_5$, and UMCI$_7$. Since the vanilla implementation sequentially executes each counterfactual inference with different input variations, the computational overhead scales approximately linearly, resulting in $2.96\times$, $5.01\times$, and $6.68\times$ the base model's inference time, respectively. We then apply a straightforward acceleration technique, called batching inference to improve the efficiency. Since each counterfactual input variations together with the original input can be executed in the forward pass independently, we can put them into one batch and conduct batch parallel acceleration. The efficiency improvement after applying batch inference is significant, the computational overhead of UMCI$_3$, UMCI$_5$, and UMCI$_7$ become $1.29\times$, $1.81\times$, and $2.48\times$, respectively. In future work, we believe that we can use KV cache sharing to further accelerate the UMCI. Since each counterfactual input modifies only either the textual or visual modality, we can exploit shared components to reduce redundant calculations. For example, when the visual input is fixed and only textual prompts vary, we can prefill the visual tokens once and reuse the KV cache across all textual variations. While this approach requires additional engineering effort and potentially model fine-tuning, it offers significant theoretical efficiency gains.

**The complete experiments on Bias/Sensitive/BS Subsets.** Due to space constraints, the original paper only presented partial results for the Bias/Sensitive/BS Subsets experiments. The complete results are provided in Table 9. Experiments on all counterfactual inference settings with variant inputs are also included. Although LLaVA-NeXT shows 0.0 accuracy on the Bias Subset, as discussed in the main paper, variants such as LLaVA-NeXT-VCF-Color0, LLaVA-NeXT-VCF-Noise400, and LLaVA-NeXT-VCF-Noise500 may still achieve non-zero performance. This is because the Bias Subset is constructed from the combination of LLaVA-NeXT-VCF-Color0 and LLaVA-NeXT-VCF-Noise500 under our proposed setting. An incorrect prediction from one variant may coincidentally be correct in another (yet, it's still a blind guess), allowing for occasional non-zero accuracies in these counterfactual settings.

| Method | Bias Subset | | | Sensitivity Subset | | | BS Subset | | |
|---|---|---|---|---|---|---|---|---|---|
| | MCQ | Others | Overall | MCQ | Others | Overall | MCQ | Others | Overall |
| LLaVA-NeXT | 0.0 | 0.0 | 0.0 | 39.2 | 37.63 | 38.63 | 15.91 | 27.58 | 18.75 |
| LLaVA-NeXT-TCF-V1 | 3.20 | 6.38 | 3.71 | 36.62 | 26.80 | 33.02 | 14.86 | 19.65 | 16.02 |
| LLaVA-NeXT-TCF-V2 | 5.80 | 8.70 | 6.26 | 24.08 | 33.51 | 27.54 | 9.77 | 24.56 | 13.36 |
| LLaVA-NeXT-TCF-V3 | 3.09 | 3.19 | 3.11 | 38.61 | 34.54 | 37.11 | 15.99 | 25.31 | 18.26 |
| LLaVA-NeXT-VCF-Color0 | 4.59 | 4.06 | 4.50 | 27.26 | 23.54 | 25.90 | 13.85 | 18.01 | 14.86 |
| LLaVA-NeXT-VCF-Noise400 | 6.63 | 3.19 | 6.08 | 27.96 | 23.71 | 26.40 | 14.98 | 17.51 | 15.60 |
| LLaVA-NeXT-VCF-Noise500 | 6.30 | 3.48 | 5.85 | 27.16 | 23.02 | 25.65 | 14.54 | 17.63 | 15.29 |
| LLaVA-NeXT-TIE | 12.98 | 23.48 | 14.66 | 39.00 | 57.56 | 45.81 | 21.89 | 44.21 | 27.31 |
| LLaVA-NeXT-VCD | 12.65 | 25.51 | 14.71 | 40.50 | 56.53 | 46.38 | 22.54 | 44.58 | 27.89 |
| LLaVA-NeXT-M3ID | 16.91 | 25.22 | 18.24 | 39.90 | 56.36 | 45.94 | 24.15 | 44.33 | 29.05 |
| LLaVA-NeXT-UMCI$_3$ (ours) | 21.22 | 35.36 | 23.48 | 39.60 | 60.31 | 47.20 | 27.14 | 50.13 | 32.72 |
| LLaVA-NeXT-UMCI$_5$ (ours) | 23.81 | 37.97 | 26.08 | **40.60** | **60.65** | **47.95** | 28.80 | 51.01 | 34.19 |
| LLaVA-NeXT-UMCI$_7$ (ours) | **24.86** | **38.26** | **27.01** | 40.10 | **60.65** | 47.64 | **29.68** | **51.26** | **34.92** |
| Qwen2-VL | 5.37 | 8.56 | 6.11 | 38.10 | 34.41 | 36.06 | 10.78 | 23.59 | 14.52 |
| Qwen2-VL-TCF-V1 | 6.11 | 11.31 | 7.32 | 36.51 | 36.01 | 36.23 | 10.38 | 24.37 | 14.46 |
| Qwen2-VL-TCF-V2 | 7.59 | 15.90 | 9.52 | 40.87 | 34.41 | 37.3 | 12.07 | 23.00 | 15.26 |
| Qwen2-VL-TCF-V3 | 6.30 | 8.87 | 6.89 | 37.70 | 34.41 | 35.88 | 11.02 | 22.42 | 14.35 |
| Qwen2-VL-VCF-Color0 | 5.83 | 6.73 | 6.04 | 20.24 | 28.94 | 25.04 | 8.77 | 18.52 | 11.62 |
| Qwen2-VL-VCF-Noise400 | 7.59 | 21.41 | 10.80 | 21.03 | 25.72 | 23.62 | 10.22 | 24.17 | 14.29 |
| Qwen2-VL-VCF-Noise500 | 7.59 | 21.71 | 10.87 | 20.63 | 27.33 | 24.33 | 10.62 | 25.15 | 14.86 |
| Qwen2-VL-TIE | 16.20 | 16.82 | 16.35 | 45.63 | 36.66 | 40.67 | 20.27 | 27.29 | 22.32 |
| Qwen2-VL-VCD | 15.74 | 21.71 | 17.13 | 46.83 | 40.84 | 43.52 | 20.11 | 30.41 | 23.12 |
| Qwen2-VL-M3ID | 19.81 | 21.71 | 20.26 | **47.22** | 41.16 | 43.87 | 23.65 | 30.6 | 25.68 |
| Qwen2-VL-UMCI$_3$ (ours) | 21.67 | 26.30 | 22.74 | 44.05 | 42.44 | 43.16 | 24.54 | 32.75 | 26.94 |
| Qwen2-VL-UMCI$_5$ (ours) | 24.91 | 25.69 | 25.09 | **47.22** | 42.44 | 44.58 | 28.00 | 33.14 | 29.50 |
| Qwen2-VL-UMCI$_7$ (ours) | **27.04** | **29.66** | **27.65** | **47.22** | **45.98** | **46.54** | **29.61** | **36.84** | **31.72** |

Table 9: The complete experiments on Bias Subset, Sensitivity Subset, and BS Subset across two widely used base LVLMs demonstrate the effectiveness of the proposed UMCI framework. **Bold texts** indicate the best result of each column.

| Method | Single Dataset | | | | | | Gathered by Question Type | | |
|---|---|---|---|---|---|---|---|---|---|
| | MMB-C | MMB-E | MME | CCB | MMS | ViLP | MCQ | Others | Overall |
| LLaVA-NeXT | 78.0 | 79.72 | 79.57 | 47.0 | 44.75 | 51.53 | 70.12 | 71.86 | 70.46 |
| LLaVA-NeXT-TC-V1 | 77.46 | 79.95 | 76.20 | 46.75 | 43.92 | 51.53 | 69.87 | 69.42 | 69.78 |
| LLaVA-NeXT-TC-V2 | 77.44 | 77.51 | 78.78 | 46.20 | 42.08 | 50.14 | 68.68 | 70.90 | 69.12 |
| LLaVA-NeXT-VC-C0 | 29.97 | 31.85 | 50.29 | 27.02 | 25.08 | 28.47 | 29.66 | 44.29 | 32.55 |
| LLaVA-NeXT-VC-N500 | 30.69 | 33.08 | 48.29 | 28.25 | 25.0 | 29.03 | 30.55 | 42.99 | 33.01 |
| LLaVA-NeXT-TIE | 78.28 | 80.28 | 77.30 | 45.65 | 46.00 | 53.19 | 70.36 | 70.68 | 70.42 |
| LLaVA-NeXT-VCD | 78.38 | 80.28 | 78.09 | 46.63 | 45.00 | 54.31 | 70.44 | 71.55 | 70.66 |
| LLaVA-NeXT-M3ID | 78.31 | 80.18 | 78.62 | 45.89 | 45.92 | 54.03 | 70.36 | 71.86 | 70.66 |
| LLaVA-NeXT-UMCI$_5$ (ours) | 78.21 | 80.08 | 80.15 | 46.20 | 45.75 | 53.06 | 70.32 | 72.70 | 70.79 |
| Qwen2-VL | 85.26 | 86.36 | 87.89 | 73.22 | 59.50 | 56.53 | 80.91 | 79.27 | 80.58 |
| Qwen2-VL-TC-V1 | 85.28 | 86.11 | 87.79 | 73.18 | 59.73 | 58.09 | 80.84 | 79.63 | 80.60 |
| Qwen2-VL-TC-V2 | 85.26 | 86.39 | 87.96 | 72.92 | 59.53 | 56.37 | 80.88 | 79.27 | 80.56 |
| Qwen2-VL-VC-C0 | 34.46 | 35.54 | 50.45 | 25.37 | 27.33 | 24.72 | 32.66 | 43.38 | 34.77 |
| Qwen2-VL-VC-N500 | 31.33 | 31.82 | 50.13 | 25.43 | 28.50 | 26.81 | 30.29 | 43.72 | 32.94 |
| Qwen2-VL-TIE | 86.00 | 86.59 | 86.52 | 73.84 | 59.00 | 57.08 | 81.30 | 78.43 | 80.73 |
| Qwen2-VL-VCD | 86.05 | 86.56 | 86.41 | 73.77 | 60.08 | 57.92 | 81.42 | 78.58 | 80.86 |
| Qwen2-VL-M3ID | 85.69 | 86.46 | 86.10 | 73.96 | 59.75 | 57.78 | 81.25 | 78.31 | 80.67 |
| Qwen2-VL-UMCI$_5$ (ours) | 85.97 | 86.67 | 87.36 | 73.59 | 59.92 | 58.06 | 81.39 | 79.31 | 80.98 |

Table 10: Experiments on MMB(ench-Dev)-C/E(N-V11), MME, CCB(ench), MMS(tar), and ViLP including all counterfactual inference results used by UMCI$_5$. Blue texts indicate an improvement over the baseline.

| Original Prompts | TC-V1 Prompts |
|---|---|
| Please select the correct answer from the options above. | Think about the question based on details in the given image. Please select the correct answer from the options above. |
| Please answer yes or no. | Think about the question based on details in the given image. Please answer yes or no. |
| Please try to answer the question with short words or phrases if possible. | Think about the question based on details in the given image. Please try to answer the question with short words or phrases if possible. |
| Answer the question directly using a single word or phrase. | Think about the question based on details in the given image. Answer the question directly using a single word or phrase. |
| Answer with the option's letter from the given choices directly. | Think about the question based on details in the given image. Answer with the option's letter from the given choices directly. |
| (Chinese Prompts) 请直接回答选项字母。 | (Chinese Prompts) 结合问题与选项仔细观察图像中的信息，请直接回答选项字母。 |

Figure 3: The list of all TC-V1 prompts that add an additional system prompt instructing the model to focus on image details.

| Original Prompts | TC-V2 Prompts |
|---|---|
| Please select the correct answer from the options above. | (Chinese Prompts) 请仔细观察图像中的信息，然后结合问题与选项，从上述所有选项中直接回答正确选项对应的字母。 |
| Please answer yes or no. | (Chinese Prompts) 观察给出的图片，请直接回答yes或no。 |
| Please try to answer the question with short words or phrases if possible. | (Chinese Prompts) 请仔细观察图像中的细节，然后结合图像上的信息回答问题，请直接用一个简短的英语单词或数字回答。 |
| Answer the question directly using a single word or phrase. | (Chinese Prompts) 请仔细观察图像中的细节，然后结合图像上的信息回答问题，请直接用一个简短的英语单词或数字回答。 |
| Answer with the option's letter from the given choices directly. | (Chinese Prompts) 请仔细观察图像中的信息，然后结合问题与选项，从上述所有选项中直接回答正确选项对应的字母。 |
| (Chinese Prompts) 请直接回答选项字母。 | Please carefully examine the information in the image, then consider the question and options, and reply directly with the letter corresponding to the correct answer from the options above. |

Figure 4: The list of all TC-V2 prompts that further modify the system prompt's language from English to Chinese or vice versa.

| Original Prompts | TC-V3 Prompts |
|---|---|
| Please select the correct answer from the options above. | You are a smart student who is good at answering multiple-choice questions. Please select the correct answer from the options above. |
| Please answer yes or no. | You are a smart student who is good at answering yes or no questions. Please answer yes or no. |
| Please try to answer the question with short words or phrases if possible. | You are a smart student who is good at answering questions. Please try to answer the question with short words or phrases if possible. |
| Answer the question directly using a single word or phrase. | You are a smart student who is good at answering questions. Answer the question directly using a single word or phrase. |
| Answer with the option's letter from the given choices directly. | You are a smart student who is good at answering multiple-choice questions. Answer with the option's letter from the given choices directly. |
| (Chinese Prompts) 请直接回答选项字母。 | (Chinese Prompts) 你是一名擅长回答选择题的聪明学生，请直接回答选项字母。 |

Figure 5: The list of all TC-V3 prompts that inject identity information by prompting the model to respond as a clever student.

