# OpenReview forum: "UMCI: A Unified Counterfactual Framework for Robust Vision-Language Reasoning"
_ICLR.cc/2026/Conference — ICLR 2026 Conference Withdrawn Submission_

### Official Review · Reviewer_RF2j · 2025-10-26

**Soundness:** 2
**Presentation:** 2
**Contribution:** 2
**Rating:** 4
**Confidence:** 2

**Summary:**

The paper proposes UMCI, a test-time method for large vision-language models (LVLMs) that unifies visual and textual counterfactual reasoning. It generalizes Visual Contrastive Decoding (VCD) by averaging logits over multiple perturbed images and paraphrased prompts to reduce language bias and sensitivity. The authors also introduce a Bias & Sensitivity Benchmark, adaptively identifying model-specific fragile samples, and report modest robustness gains on this benchmark and standard datasets.

**Strengths:**

* This paper provides a unifying perspective linking VCD to causal debiasing (TIE/TDE) and interprets temperature as a causal weighting mechanism.
* UMCI is simple to apply at inference and it includes prior methods as special cases (VCD ≈ VC only; CF-VQA ≈ TIE only). The decomposition into VC and TC is intuitive.
* The proposed benchmark formalizes bias and sensitivity via explicit criteria and highlights that non-robust samples vary across LVLMs, which is a useful diagnostic perspective.

**Weaknesses:**

* Minor performance gains. In Table 4, the improvements on standard benchmarks are very small, often within noise. Most gains appear only on the BS Benchmark, which is constructed using the model’s own failure cases.
* The method mainly combines known ideas — visual perturbations and prompt ensembling — under a unified formulation. The innovation over prior works (VCD, TIE, TDE) is modest.
* UMCI uses more test-time compute (multiple counterfactual rounds) than baselines. Compute-matched ensemble baselines are not compared.

**Questions:**

na

---

### Official Review · Reviewer_N6gF · 2025-10-29

**Soundness:** 3
**Presentation:** 3
**Contribution:** 3
**Rating:** 4
**Confidence:** 3

**Summary:**

The authors identify two key robustness challenges in current Vision Large Models (VLMs): a bias towards prioritizing language over vision, and a high sensitivity to prompts. To address these challenges, they propose the Unified Multi-round Counterfactual Inference (UMCI) framework. This framework adjusts the input to generate multi-round results and performs visual and linguistic debiasing at the logit level. Concurrently, the authors introduce a method to dynamically construct a benchmark (the BS benchmark) designed to measure the bias and sensitivity of different inference methods for a specific model.

**Strengths:**

1. The paper is well-written with a clear and easy-to-follow logical flow.
2. The proposed method achieves significant improvements on the paper's BS benchmark. When evaluated on general benchmarks, it outperforms other inference methods in terms of both the number of benchmarks improved and overall performance.

**Weaknesses:**

1. The construction of the BS benchmark shares structural similarities with the proposed method. This raises a concern that the selected subset might be inherently biased, creating a situation where the proposed method is "both the player and the referee" in its own evaluation.
2. The analysis of the BS benchmark could be more in-depth. For instance, it is unclear how the resulting subsets differ when constructed from different sets of visuals (v) and questions (q), and whether this would lead to different evaluation results.
3. As described, the use case for the BS benchmark appears limited. It evaluates the relative bias and sensitivity of different inference methods for the same model. In practice, the community is often more concerned with the absolute bias and sensitivity exhibited by a specific model in application.
4. The paper suggests that Visual Counterfactual (VC) primarily mitigates the bias of neglecting vision (corresponding to the B subset), while Textual Counterfactual (TC) mainly reduces sensitivity to text prompts (corresponding to the S subset). It is not clear if this correspondence is explicitly reflected in the metrics.
5. In Table 5, the contribution of TC appears to be limited, with VC playing the primary role. Their respective contributions on the general benchmarks are not clearly delineated.
6. The generalizability of TC may be a concern, as it seems to require specific designs for specific problems. Its application in more general scenarios needs further analysis.
7. The proposed method shows substantial gains on the BS subset but smaller improvements on general benchmarks. This raises the question of whether the method might degrade performance on the remaining parts of the full test set.
8. Line 51 of the paper states that language bias has some overlap and connection with hallucination. Given this, the performance of the proposed method, the baseline, and the main compared methods on a benchmark like HallusionBench should be presented.
9. Regarding Equation 5, it is unclear if it represents a token-level probability distribution. If so, since VLMs generate text autoregressively, the textual condition for generating each token should be the prompt 'q' combined with the previously generated tokens, not just 'q'.

**Questions:**

See Weaknesses.

If most of my concerns can be well addressed, I would like to raise my score.

---

### Official Review · Reviewer_wSys · 2025-10-31

**Soundness:** 3
**Presentation:** 3
**Contribution:** 2
**Rating:** 4
**Confidence:** 4

**Summary:**

This paper presents UMCI (Unified Multi-round Counterfactual Inference), an inference-time framework aimed at improving the robustness of large vision–language models (LVLMs).UMCI unifies visual and textual counterfactual reasoning within a single process: the model performs multiple rounds of counterfactual queries and aggregates the resulting logits to produce more stable predictions.
Building on previous causal inference methods such as TIE, VCD, and M3ID, the authors extend these approaches by incorporating textual counterfactuals generated through simple templates—covering tone variation, language switching, and role prompting.The framework is evaluated on two representative LVLMs, LLaVA-NeXT and Qwen2-VL, across multiple multimodal benchmarks.Results show small yet consistent gains in robustness. The paper also provides detailed experimental settings and introduces a dynamic benchmark to test sensitivity to linguistic and visual perturbations under realistic conditions.
In summary, UMCI offers:

1）A unified, multi-round inference framework for enhancing LVLM robustness.

2）Joint treatment of visual and textual counterfactuals within a causal reasoning paradigm.

3）Transparent and reproducible experiments on two distinct LVLM architectures.

4）A discussion of inference-time efficiency and robustness trade-offs.

**Strengths:**

1）Clear Motivation and Unified Framework
The paper addresses an important yet underexplored problem—how to improve the inference-time robustness of large vision–language models (LVLMs) without retraining. By unifying visual and textual counterfactual reasoning under a single causal formulation, UMCI offers a coherent and conceptually clear framework that extends prior causal decoding approaches such as TIE, VCD, and M3ID. This unified perspective helps bridge previously separate research lines in visual and linguistic robustness.

2）Transparent and Reproducible Experiments
The experiments are fully transparent, with detailed descriptions of datasets, model architectures, and all hyperparameters. The study relies entirely on open-source LVLMs (LLaVA-NeXT and Qwen2-VL) and public toolkits, ensuring that all reported results are easily reproducible and verifiable. Such experimental rigor adds credibility and makes the framework accessible for future benchmarking.

3）Cross-Model Generalization
The evaluation covers two representative LVLMs—LLaVA-NeXT, which is more vision-focused, and Qwen2-VL, which has stronger language grounding. UMCI demonstrates consistent gains across these different architectures, indicating that the method is not tied to a specific model type. This cross-model validation strengthens the claim of generality and suggests potential applicability to broader multimodal systems.

4）Strong Engineering Integration
The framework integrates multiple existing components—visual counterfactuals, textual counterfactuals, and multi-round inference—into a unified and reusable pipeline. This design reflects strong engineering execution and makes the framework practically useful. Researchers can readily adapt the system for robustness studies, counterfactual evaluation, or future model comparisons.

5）Dynamic Robustness Evaluation Benchmark
The proposed dynamic benchmark supports model-adaptive robustness evaluation under both visual and linguistic perturbations. Unlike traditional static bias datasets, it allows more flexible and realistic testing that better reflects real-world input variations. This makes the benchmark itself a valuable tool for advancing robustness evaluation practices in multimodal research.

**Weaknesses:**

1）Limited Methodological Novelty
The proposed UMCI framework primarily integrates existing causal inference approaches—TIE, VCD, and M3ID—into a single formulation. While this unification is conceptually coherent, it introduces limited algorithmic innovation. Both the visual counterfactuals (black, blurred, or noisy images) and the textual counterfactuals (template-based rewrites) largely follow existing methods or rely on simple heuristics. Consequently, the contribution is more of an engineering consolidation than a genuine methodological advancement.

2）High Inference Cost and Practical Constraints
UMCI requires multiple inference rounds (typically three to seven) to achieve stable results. This increases inference latency by roughly 1.8×–2.5×, alongside proportional growth in token usage and GPU memory consumption. Although batch inference can partially offset the delay, the memory footprint still scales linearly with the number of rounds, limiting feasibility on consumer GPUs or edge devices. Moreover, the paper lacks quantitative measurements of token-level overhead or throughput degradation, leaving its practicality for real-time applications uncertain.

3）Static and Simplistic Textual Counterfactual Design
The textual counterfactual module is based on three fixed templates—tone modification, language switching, and role prompting—which are handcrafted and deterministic. This design lacks semantic diversity and adaptivity, making it unable to capture the broader range of linguistic sensitivity cases. While the authors claim these represent typical scenarios, they provide no statistical evidence or empirical justification. As a result, the textual component feels simplistic and may not generalize beyond the tested benchmarks.

4）Marginal Performance Improvements
The reported gains across standard benchmarks are modest, typically below 0.5%, and in some cases even lower than those achieved by earlier methods such as M3ID. Considering the 3–7× inference repetition, the trade-off between robustness and efficiency remains weak. Furthermore, the “scaling law” analysis (Section 4.4) shows performance saturation after five rounds and lacks statistical validation, suggesting that the claimed improvement trend may not be robust.

5）Outdated Baselines and Missing Comparisons
Although UMCI compares against several causal inference and decoding-based baselines (TIE 2021, VCD 2024, M3ID 2024), it omits more recent methods that have advanced inference-time robustness for LVLMs. For example, LCD (ACL 2024), ICD (ACL 2024), and RVCD (ACL 2025) extend contrastive decoding and causal reasoning with improved robustness and efficiency. Without experiments or discussion involving these newer methods, it is difficult to determine whether UMCI truly advances the state of the art. As a result, the paper feels somewhat dated and incomplete in benchmarking, which weakens the strength of its empirical claims.

**Questions:**

1）Computational Cost and Resource Usage
Could the authors provide detailed measurements of token usage and GPU memory consumption for different inference rounds (e.g., UMCI₃, UMCI₅, UMCI₇)?Including such data would give a more comprehensive understanding of the computational cost beyond latency alone and clarify the framework’s scalability in practical settings.

2）Textual Counterfactual Categorization
The paper divides textual counterfactuals into three categories, but this classification appears heuristic and lacks empirical or systematic justification.Could the authors provide evidence or statistical analysis showing that these three types adequately represent major linguistic sensitivity patterns?Additionally, since fixed templates may not generalize well across diverse prompts, have the authors considered using dynamic or adaptive counterfactual generation methods?

3）Robustness–Accuracy Trade-off
UMCI shows strong gains on the BS Benchmark but only marginal or even negative improvements on standard benchmarks.Does this suggest that the robustness enhancement comes at the cost of reasoning accuracy?It would be helpful if the authors could clarify this trade-off and discuss possible ways to mitigate performance degradation on clean data.

4）Cross-Model Inconsistency
UMCI exhibits inconsistent performance across models: it yields larger gains on Qwen2-VL but very limited or even negative changes on LLaVA-NeXT across several datasets.This discrepancy implies that UMCI’s effectiveness may depend on specific model architectures or training alignments rather than being universally applicable.Could the authors elaborate on the reasons behind this difference?Does UMCI require model-specific tuning to maintain consistent improvements across architectures?

---

### Official Review · Reviewer_xDzz · 2025-11-03

**Soundness:** 2
**Presentation:** 3
**Contribution:** 3
**Rating:** 6
**Confidence:** 3

**Summary:**

The paper presents a systematic study of visual contrastive decoding (VCD; Leng et al, 2024) in context of previous counterfactual inference methods such as CF-VQA (Niu et al, 2021), and proposes a unified inference framework for counterfactual bias mitigation in large vision-language models. Specifically, the authors show that VCD is equivalent to reweighting the original output probabilities by the exponential of total indirect effect (TIE) logits as defined in CF-VQA, and generalize this formulation by repeating the inference and reweighting over multiple text and visual counterfactuals. Experiments on a new bias-sensitivity benchmark demonstrate this multi-round inference procedure reduces language bias and improves robustness to text perturbations.

**Strengths:**

- Neat, systematic approach to unify existing methods on counterfactual inference of LVLMs. UMCI treats CF-VQA and VCD as special cases but bridge and generalize them to enable more diverse counterfactuals, reducing inconsistencies of the output.
- Interesting method to probes bias and sensitivity of models by bootstrapping from existing benchmark data. This allows measuring and comparing bias and robustness of LVLMs in realistic settings, not artificial tasks from previous bias/hallucination benchmarks.
- UMCI shows promising results on the bias-sensitivity benchmark, outperforming baselines by large margins and demonstrating some degree of scaling over inference rounds.

**Weaknesses:**

- The counterfactuals in the proposed BS benchmark are generated using the same procedures as in the proposed UMCI method. This seems to lead to exaggerated improvements over the baselines (table 2), while their performances are much closer on real-world benchmarks (table 4). In other words, it is not clear to me how well the method generalizes beyond the counterfactual types used at inference time (a crucial dimension of test-time scaling in my opinion).

- While the definition of the BS benchmark makes sense for MCQ and binary questions, I'm not sure it is suitable for open-ended tasks when the output is more than one or a few words (ViLP), as it is highly unlikely the model generates identical long responses over multiple runs, especially under nondeterministic sampling. I wonder if using generative evaluation (GPT judge) or some form of semantic matching like https://arxiv.org/abs/2302.09664 may make the benchmark more robust for true open-ended QA?

- Test-time scaling seems to saturate after 3-5 inferences, more apparent for LLaVA-NeXT. While the general trend is still positive, I would be hesitant to consider it a definite proof of "scaling law" (in asymptotic sense) before at least experimenting with more models and sampled counterfactuals.

**Questions:**

See weaknesses. Also, are the optimal hyperparameters ($\tau_1$, $\tau_2$) similar across models? Is it possible to use different numbers of M and N and study scaling behavior along both directions?

---

### Note · Authors · 2025-11-13

I have read and agree with the venue's withdrawal policy on behalf of myself and my co-authors.